WASH regulates the oxidative stress Nrf2/ARE pathway to inhibit proliferation and promote apoptosis of HeLa cells under the action of Jolkinolide B

Hong Yu 1
Liu Jicheng 1
Kong Wanying 1
Li Hui 1
Cui Ying 1
Liu Yuchao 1 2
Deng Zhihui 1
Ma Dezhi 1
Zhang Keyong 1
Li Jinghui 1 ljh20120525@126.com
Li Minhui 1 2 3 prof_liminhui@yeah.net
1 Qiqihaer Medical University , Qiqihaer , China
2 Baotou Medical College , Baotou , China
3 Inner Mongolia Hospital of Traditional Chinese Medicine , Hohhot , China
Paranjape Anurag
Electronic publication date: 2022 Jul 13
Publication date: 2022
Volume: 10
Electronic Location ID: e13499
Received 2021 Dec 21; Accepted 2022 May 5
Copyright: © 2022 Hong et al.
Copyright year: 2022
Copyright holder: Hong et al.
License: This is an open access article distributed under the terms of the Creative Commons Attribution License, which permits unrestricted use, distribution, reproduction and adaptation in any medium and for any purpose provided that it is properly attributed. For attribution, the original author(s), title, publication source (PeerJ) and either DOI or URL of the article must be cited.
License URL: https://creativecommons.org/licenses/by/4.0/

Keywords: JB, Apoptosis, Cervical cancer, Nrf2/ARE pathway, WASH protein

Funding: Science and Technology Plan Joint Guidance Project, Qiqihar LHYD-2021013 Science and Technology Innovation Guidance Project, Inner Mongolia KCBJ2018040 This work was supported by the Science and Technology Plan Joint Guidance Project, Qiqihar (No. LHYD-2021013), and the Science and Technology Innovation Guidance Project, Inner Mongolia (No. KCBJ2018040). There was no additional external funding received for this study. The funders had no role in study design, data collection and analysis, decision to publish, or preparation of the manuscript.

==============================
Jolkinolide B (JB), a diterpenoid compound isolated from the roots of Euphorbia fischeriana, has gained research attention for its antitumor effects. In recent years, JB reportedly displayed anti-tumor activity in solid tumors, such as breast, ovarian, and prostate cancer, and leukemia. In this study, we evaluated the effect of JB on HeLa cells with a focus on cell growth inhibition and related mechanisms. HeLa cells were cultured in vitro and divided into a blank control group, HeLa-Scramble (0, 0.25, 0.5 mM), and Wiskott-Aldrich syndrome protein and SCAR homolog (WASH) protein silenced group, HeLa-shWASH (0, 0.25, 0.5 mM). Morphological changes were observed using an inverted microscope. The inhibition rate of cell proliferation was detected using the WST-1 method. Flow cytometry Brdu+PI double standard method was used to detect cell replication ability and FITC+PI double standard method was used to detect cell apoptosis rate. Western blot was used to verify the expression of Nrf2, HO-1, WASH, Bax, Bcl-2, and PCNA. The mRNA expression of cytokines (IL-1α, IL-6, and IL-8) was detected using RT-qPCR. The results showed that JB induced cell apoptosis and arrested cells at the G2/M phase in HeLa-shWASH cells compared with HeLa-Scramble cells (P < 0.05, P < 0.01, respectively). In addition, JB upregulated IL-1α, IL-6, and IL-8 in HeLa-shWASH cells. We conclude that WASH protein participates in JB-induced regulation of the Nrf2/ARE pathway, aggravates inflammatory responses, and promotes cancer cell apoptosis, thus inhibiting the proliferation and invasion abilities of HeLa cells. JB may have anti-tumor effects and potential clinical value for the treatment of cervical cancer.

Introduction

Cancer is a deadly disease with high mortality rates in humans and is a growing public health concern. The incidence and mortality of cancer are increasing each year. Cervical cancer is a malignant tumor that poses a threat to women’s health worldwide. In recent years, with an increase in human papillomavirus (HPV) infections, the incidence of cervical cancer in young women (under the age of 35) has increased and is now one of the leading causes of cancer deaths in women (Parkhurst & Vulimiri, 2013). Current clinical programs for the treatment of cervical cancer often cause adverse reactions or serious complications. In addition, many commonly used anticancer chemotherapeutics have a certain degree of toxicity to normal cells. As such, there is an urgent need to develop effective, non-cytotoxic chemotherapeutics (Mackay, Wenzel & Mileshkin, 2015), and to discover new anti-cancer strategies for patients with cervical cancer.

The medicinal properties of plants have been recognized by many cultures for thousands of years. The application of traditional treatments has attracted widespread attention in recent years. Many medicinal plants have been applied as cancer treatments in human clinical trials and have showed promising anticancer effects due to their hypotoxicity, security, and general availability (Aiello et al., 2019). Modern medicine and traditional Chinese medicine studies have found that Euphorbia fischeriana displays anti-tumor effects and is clinically used to treat lung cancer and malignant tumors of the digestive system. Various monomer compounds extracted from Euphorbia fischeriana have demonstrated good therapeutic effects on various malignant tumors and leukemia, and have no long-term adverse reactions, unlike chemotherapy (Choene & Motadi, 2016; Bano et al., 2017). The root of Euphorbia contains diterpenoids, triterpenoids, flavonoids, and volatile oils. Jolkinolide B (JB) is a diterpenoid extracted from the root of Euphorbia; the compound has multiple biological activities (such as anti-proliferation, anti-bacterial, and anti-viral effects) (Wang et al., 2011), a molecular formula of C20H26O4, and a molecular weight of 330.42 Da. Studies have confirmed that rock euphorbia lactone B prevents proliferation of various carcinoma cells by promoting apoptosis and inhibiting migration; moreover, it exhibits extensive anti-tumor activity in vitro and in vivo (Lin et al., 2012). Although JB can inhibit the growth of various cancer cells, it is less toxic to normal cells, and hence, it may be the first choice for cancer treatment. Early studies have shown that JB has a significant inhibitory effect on solid tumors, such as breast, ovarian, and prostate cancer, and leukemia, and can be used for the treatment of malignant tumors (Wang et al., 2013). However, it has not been reported whether JB induces apoptosis and inhibits tumor growth through other mechanisms in cervical cancer. To the best of our knowledge, other impacts, such as boosting immunity or anti-angiogenic effects in vitro have not been assessed.

The change in cytoskeleton structure and function is also an essential factor leading to tumor occurrence and development. Many studies have shown that the malignant phenotype of tumor cells is usually associated with the abnormal expression of actin-related proteins. In this study, we focused on Wiskott-Aldrich syndrome protein (WASP), which is a cytoskeleton regulatory protein, and SCAR homolog (WASH), which is an important cytoskeleton regulatory-related protein in the WASP family. Many studies have reported that cytoskeleton regulatory proteins are closely associated with tumor genesis and development (Papakonstanti & Stournaras, 2008). However, the mechanism of cytoskeleton regulation in the development, migration, and metastasis of tumor cells. Cytoskeleton regulatory proteins have notable effects on the expression of some important genes (Eberhardt et al., 2016; Gehren et al., 2015). Moreover, the cytoskeleton is closely related to many signaling pathways (Duleh & Welch, 2010; Gomez & Billadeau, 2009). It is questionable whether Nrf2/ARE pathway is involved in the mechanism of WASH protein affecting JB sensitivity to cervical cancer cells. However, the involvement of WASH in tumor development has not been reported. Whether WASH, as an important cytoskeletal regulatory protein, affects the fate, progression, and metastasis of tumor cells is worthy of further research. In this study, we explored the effect of WASH expression during exposure of cervical cancer cells to JB, analyzed the mechanism by which WASH participates in Nrf2/ARE pathway regulation, and observed its influence on the therapeutic effect of JB on cervical cancer. To this end, a silencing WASH protein model was constructed for cervical cancer cells to observe the effects of various concentrations of JB on the morphology, cell cycle, and apoptosis rate of human cervical cancer (HeLa) cells. Our results provide a theoretical basis for the clinical inclusion of JB as a new type of anticancer drug and a potential chemotherapeutic drug for human cervical cancer treatment. Our results suggest that JB has far-reaching research prospects.

Therefore, here, we explored the influence of JB on the occurrence and development of cervical cancer and further analyzed whether JB affects the progress of cervical cancer through the joint mechanism of WASH and Nrf2/ARE pathway, in order to obtain a better therapeutic target for cervical cancer.

Materials and Methods

Cell line

Human cervical cancer (HeLa) cells were purchased from the National Experimental Cell Resource Sharing Platform.

Drugs and reagents

JB was supplied by Canadian TRC with a purity of ≥98% (item number: P468400). The WST-1 cell proliferation and cytotoxicity detection kit was supplied by Biyuntian Biotechnology (C0036; Shanghai, China). The apoptosis kit was supplied by BD, USA (556547; Franklin Lakes, NJ). Anti WASH1 antibody-C-terminal (ab157592; Abcam, Cambridge, UK). Rabbit monoclonal antibody GAPDH (article number: 60004-1-lg), rabbit monoclonal antibody Nrf2 (article number: 16396-1-AP), and rabbit monoclonal antibody PCNA (article number: 10205-2-AP) were provided by Proteintech Group, Inc. (Rosemont, IL). Secondary rabbit (catalog number: sc-2357) and mouse antibodies (catalog number: sc-516102) were provided by Santa Cruz Biotechnology (Dallas, TX, USA). Dulbecco’s Modified Eagle Medium (DMEM) and fetal bovine serum (FBS) were purchased from Hyclone (Logan, UT) and CLARK Bioscience (Richmond, VA), respectively. Trypsin was purchased from Gibco (Thermo Fisher Scientific, Waltham, MA).

Laboratory apparatus

Laboratory apparatus used were as follows: Anthos Zenyth 200 full-wavelength microplate reader (British Biochrom, Cambridge, UK); IX73 fluorescence inverted microscope (OLYMPUS, Tokyo, Japan); HHR40-II-A2 biological safety cabinet and DW-86L626 ultra-low temperature refrigerator (Qingdao Haier Special Electric Co., Ltd., Qingdao, China); W2001R carbon dioxide Incubator (SIM, Charlotte, NC, USA); 4-16K high-speed refrigerated centrifuge (Sigma, Darmstadt, Germany); BDFACSVerse flow cytometer (General Electric, Schenectady, NY, USA); PowerPac Bsic electrophoresis instrument and Mini Subcell GT type horizontal electrophoresis tank (Life Technologies Company, Carlsbad, CA, USA); AI680 gel imaging system (GE, Fort Worth, TX, USA); 153BR protein transfer membrane instrument (Bio-Rad, Hercules, CA, USA); and DV215CD precision electronic balance (Ohaus, Parsippany, NJ, USA).

Cell culture

Cervical cancer (HeLa) cell line was cultured in DMEM (pH 7.2–7.4) containing 10% FBS and 1% cyclin-streptomycin double resistance at 37 °C in a 5% CO2 constant temperature incubator with saturated humidity. When cells reached 80% confluency, the passage culture was carried out.

Establishment of a cell line for WASH gene silencing mediated by shRNA lentivirus

Construction of the shRNA lentivirus expression vector targeting WASH gene

A set of shRNA sequences (shWASH: 5′-GCGCCACTGTGTTCTTCTCTA-3′, shScramble: 5′-CCTAAGGTTAAGTCGCCCTCG-3′) specific to the human WASH gene and control shRNA sequences were designed according to WASH gene sequencing using small interfering RNA online design software. Double-stranded DNA was connected to the vector by DNA ligase, and the positive clones were screened out and identified by sequencing after transformation by Stable3 receptive cells.

Construction of stable WASH-knockdown cell lines

The 293T cells were transfected with PEI as the transfection reagent; the lentivirus plasmid and package plasmid were co-transfected into 293T cells. After transfection for 4 h, fresh culture medium was substituted. Approximately 48 h after transfection, the cell supernatant containing lentivirus was collected and filtered through a 0.45-μm filter. HeLa cells in the logarithmic growth phase were inoculated into six-well plates and cultured in a 5% CO2 incubator at 37 °C until cell adhesion occurred. Then, 1 mL of virus suspension and polybrene with a final concentration of 8 mg/L were added. After 24 h of lentivirus infection, the high-glucose DMEM containing 2 mg/L puromycin was used for screening, and stable cell lines were obtained after ~7 days of screening.

WASH protein expression in stable cell lines detected using western blot

Stable cells in each group were collected, and the total protein was isolated. After protein quantification, 40 μg of WASH protein was sampled for SDS-PAGE. After membrane transfer, GAPDH was used as an internal reference to analyze expression changes of target bands.

Morphological cell changes observed under inverted microscope

Cells were removed at the logarithmic growth stage using ~1 mL trypsin for 3 min, followed by digestion termination. The supernatant was centrifuged, and the cells were suspended in serum-containing medium. The cells were inoculated into a six-well plate with ~5 × 104 cells per well and cultured at 37 °C with 5% CO2 for 24 h. The culture medium was removed, and the JB DMEM was added to the Scramble and shWASH groups to achieve final concentrations of 0.25 and 0.5 mM in each group, respectively. The same volume of culture medium was added to a blank control group. The influence of various concentrations of JB on cell state was observed after 12 h culture.

Inhibition rate of cell proliferation detected using the WST-1 method

WST-1 is a widely recognized method for the detection of tumor cell proliferation and toxicity. The experimental design was divided into a blank control group (HeLa-Scramble) and HeLa-shWASH groups (0, 0.25, and 0.5 mM); each group had two multiple holes. Cells were removed using trypsin for 2 min and centrifuged. After digestion was terminated in DMEM, the cells were counted and inoculated into 96-well plates with 5 × 103 cells per well and a final volume of 100 μL per well. After incubation at 37 °C with 5% CO2 for 24 h, 100 μL of DMEM containing JB was added to the final concentration of JB at 0, 0.25, and 0.5 mM, respectively. Culture was continued at 37 °C with 5% CO2. We then added 10 μL WST-1 solution to each well for 6, 12, 24, 36, and 48 h, and continued to incubate for 2 h. Absorbance was measured at 450 nm using a microplate reader.

Cell proliferation ability detected using BrdU

BrdU is a thymine nucleoside analog that can be incorporated into the replicating DNA molecule during cell proliferation to quickly detect the DNA replication and cell proliferation capacities. Cells were removed and inoculated into six-well plates, with the number of cells being 5 × 104 per well. The cells were cultured in complete DMEM at 37 °C under 5% CO2 for 24 h, and then treated with culture medium at a final concentration of 0, 0.25, or 0.5 mM JB for 12 h. We performed the BrdU experiment according to the manufacturer’s instructions.

Annexin V-FITC/PI double staining method for cells apoptosis

Cells with good growth condition were inoculated into six-well plates (1 × 105 cells per 1 mL and 2.5 mL per well). The cells were incubated at 37 °C for 24 h in a 5% CO2 incubator until completely adhered to the culture vessel. Thereafter, JB (0, 0.25, and 0.5 mM) was added to the cells and incubated for 12 h under the same conditions. Next, cells were collected and washed twice with phosphate-buffered saline (PBS), which was pre-cooled overnight at 4 °C. Annexin V-FITC (5 μL) and PI (5 μL) were added successively and mixed with cells, followed by incubation at room temperature for 30 min in the dark. Binding buffer (400 μL) was added to each tube, and up-flow cytometry was used for analysis within 1 h.

Cell cycle detected using flow cytometry

HeLa cells were removed using trypsin and inoculated into six-well plates at a density of 5 × 105 cells per well. The complete DMEM was added, and cells were incubated at 37 °C in a 5% CO2 incubator for 24 h. JB (0, 0.25, and 0.5 mM) was added. The blank control group received the same volume of medium and was incubated for 48 h under the same conditions. Thereafter, the culture medium was discarded. Approximately 1 × 106 cells were collected in a centrifuge tube and washed twice with PBS. After fixation with 70% ethanol at 4 °C, the fixative was removed by centrifugation and the cells were washed twice with PBS. The PBS solution was discarded, and 500 μL PI/RNaseA staining solution was added before cells were incubated at room temperature under dark conditions for 60 min. Cell cycle distribution was detected using flow cytometry.

Expression of related proteins detected using western blot

The cells were seeded into six-well plates (1 × 105 cells per 1 mL and 2.5 mL per well). Completely adherent cells were incubated at 37 °C in a 5% CO2 incubator for 24 h, JB (0, 0.25, and 0.5 mM) was added, and cells were removed after incubation for 12 h. The culture medium was discarded, and cells were washed twice with pre-cooled PBS. The cells were placed on ice, and the NP-40 lysate containing protease inhibitor was added for static lysis for 30 min. Cells were collected with a cell scraper, centrifuged at 3,000 r·min−1 at 4 °C for 3 min, and the protein concentration of the supernatant was determined using a bicinchoninic acid (BCA) protein quantitative kit. We then added 6× the corresponding volume of loading buffer and boiled at 95 °C for 5 min to obtain protein samples. After an appropriate volume of protein was removed using a pipette, the electrophoresis apparatus was operated at a constant voltage of 80 V for 30 min, and proteins were separated according to size. Electrophoresis was conducted at a constant pressure of 100 V until there was 1.5 cm at the bottom of the electrophoresis tank, and proteins were transferred to polyvinylidene fluoride (PVDF) membranes. After blocking with 4% BSA for 30 min, PVDF membranes were probed using primary antibodies (GAPDH, WASH, Nrf2, HO-1, PCNA, Bax, and Bcl-2) and incubated at 4 °C overnight, washed with TBST thrice, and incubated at room temperature for 1.5 h with the secondary antibody. Membranes were washed with TBST thrice for 10 min each. After exposure, development, and recording of each strip gray value, the target protein gray value was divided by the reference protein gray value for normalization.

Cytokines detected using RT-qPCR

mRNA expression levels of IL-1α (IL-1a-R; TTAGTGCCGTGAGTTTCCC; IL-1α-F; TGTATGTGACTGCCCAAGATG), IL-6 (IL6-F; ACTCACCTCTTCAGAACGAATTG; IL6-R; CCATCTTTGGAAGGTTCAGGTTG), and IL-8 (IL8-F; ACTGAGAGTGATTGAGAGTGGAC; IL8-R; AACCCTCTGCACCCAGTTTTC) were detected. After treatment with JB concentrations (0, 0.25, and 0.5 mM), total RNA and reverse transcription cDNA were extracted according to the instruction of the total RNA extraction kit, and cDNA was used as template for RT-qPCR detection in each group. PCR primers for IL-1α, IL-6, and IL-8 genes were designed based on the Primer 5.0 software. The qPCR conditions were as follows: 50 °C for 2 min, 95 °C for 2 min, 95 °C for 15 s, 60 °C for 1 min; this was repeated for 40 cycles. The qPCR experiment was repeated three times. The relative expression levels of IL-1α, IL-6, and IL-8 were obtained using GADPH as an internal reference.

Statistical analysis

Statistical software (SPSS20.0, Graphpad 8.0, FLowjo 7.6.1, and Image J 1.8.2) were used for the data analysis. Each experiment was repeated three times. The measurement data are presented as mean ± standard deviation. The significance of differences between groups was analyzed using a T test. P < 0.05 was considered statistically significant, and P < 0.01 was considered extremely statistically significant.

Results and analysis

Establishment of WASH-silenced cell lines

Stable cells were collected, total protein was extracted, and the expression of WASH protein was detected using western blot. The gray scale ratio between WASH protein and GAPDH band could indicate the relative expression of WASH protein. The results show that compared with HeLa-Scramble, the expression of WASH protein in HeLa-shWASH gene-silenced cells was significantly down-regulated (P < 0.01; Fig. 1), indicating that the WASH gene-silenced stable cell line was successfully constructed.

Figure 1 Construction of WASH gene-silenced stabilized cell lines.

WASH protein expression decreased in cervical cancer cells after WASH protein sinking (**P < 0.01 vs control). Three groups in parallel, n = 3.

Effect of JB on morphology of HeLa cells

Observations obtained using the inverted microscope indicated that HeLa-Scramble and HeLa-shWASH cells in the non-JB group displayed a good growth state; they were spindle type with a uniform and dense distribution (Fig. 2). In the treatment group, cell necrosis increased with increasing JB concentration. In addition, increased JB sensitivity was reported after WASH protein knockout.

Figure 2 (A–F) The influence of different concentrations of JB on HeLa-shWASHa and HeLa-Scramble cells.

Inhibitory effect of JB on HeLa cell proliferation

The proliferation inhibition rate of HeLa-Scramble and HeLa-shWASH cells significantly increased after treatment for 6, 12, 24, 36, and 48 h in the JB concentration groups. JB increased the proliferation inhibition rate of HeLa-Scramble and HeLa-shWASH in a concentration-dependent manner (P < 0.05, P < 0.01). The inhibition rates of HeLa-Scramble and HeLa-shWASH cells in the JB (0.25 and 0.5 mM) groups increased appreciably (P < 0.01) in a time-dependent manner after treatment for 24, 36, and 48 h. The results show that JB inhibited HeLa cell proliferation in a concentration- and time-dependent manner (Fig. 3).

Figure 3 Effects of JB concentration and treatment time on the proliferation of HeLa-Scramble and HeLa-shWASH cells.

JB inhibited BrdU incorporation into HeLa cells

Results showed that the positive rate of BrdU in the shWASH group was significantly higher than that in the Scramble group, and with increasing JB concentration, the rate of BrdU positive cells significantly increased (P < 0.01). Results show that JB effectively suppressed the proliferation of HeLa cells in the absence of WASH protein. In addition, in the absence of WASH protein, the sensitivity of 0.25 mM JB was higher than that of 0.5 mM JB (Fig. 4).

Figure 4 Effect of JB concentration on HeLa-shWASHa cell proliferation.

BrdU positive cells significantly increased in cervical cancer cells after WASH protein sinking (**P < 0.01 vs control). Three groups in parallel, n = 3.

Effect of JB on apoptosis of HeLa cells

Apoptosis was detected using the Annexin V-FITC/PI double staining method (Fig. 5). The results show that the apoptosis rates of HeLa-Scramble and HeLa-shWASH cells in the JB concentration groups (0, 0.25, and 0.5 mM) significantly increased after 12 h treatment; the difference was statistically significant compared with the corresponding blank control group (P < 0.01). With increasing concentration, there was a significant concentration dependence, and the difference between groups was statistically significant (P < 0.01). The results show that JB can induce apoptosis in a concentration-dependent manner in the absence of WASH protein.

Figure 5 Effect of JB concentration on the apoptosis of HeLa-shWASHa cells.

Apoptosis cells significantly increased in cervical cancer cells after WASH protein sinking (*P < 0.05 and **P < 0.01 vs control). Three groups in parallel, n = 3.

Effect of JB on HeLa-shWASH cell cycle

After treatment with 0, 0.25, or 0.5 mM JB for 12 h, the proportions of G0/G1 phase cells in the HeLa-Scramble and HeLa-shWASH groups significantly decreased, while the proportions of G2/M phase cells significantly increased (Fig. 6); the differences were more obvious in the HeLa-shWASH group and were significant (P < 0.05, P < 0.01). The concentration dependence and the difference were significant (P < 0.05). The results show that JB can arrest G2/M phase cells more effectively in the absence of the WASH protein. In addition, the data showed that the number of cells in S phase increased after JB treatment, while G1 significantly decreased, indicating that JB can effectively inhibit cell proliferation, which is due to some reason that the tasks that should be completed in S phase are not completed or checkpoint problems occur. After the treatment of WST-1, it was found that the cell viability was indeed decreased by this treatment, and the apoptosis experiment showed that the cell apoptosis was also increased by this treatment, indicating that the decrease of cell viability was due to the cells were also blocked in the S phase.

Figure 6 Effect of JB concentrations on the HeLa-shWASHa cell cycle.

In the absence of WASH protein, JB effectively blocks cells during the G2/M phase. Three groups in parallel, n = 3.

Effect of JB on expression of HeLa-shWASH cell-related proteins

The effects of JB concentration on the expression of HeLa-Scramble and HeLa-shWASH anti-oxidative stress pathway, cell cycle, and apoptosis-related proteins (Nrf2, HO-1, PCNA, Bax, and Bcl-2) are shown in Fig. 7. The results showed that in the HeLa-Scramble group, the expression levels of key proteins Nrf2 and HO-1 in the Nrf2/ARE pathway increased gradually after the HeLa cells were treated with JB concentrations (0.25, and 0.5 mM) for 12 h. In the HeLa-shWASH group, the expression of Nrf2 gradually decreased and the expression of HO-1 increased. Compared with the negative control group, the difference was significant (P < 0.05). However, the HO-1 protein expression level in the HeLa-shWASH group was higher than that in the HeLa-Scramble group (P < 0.05), while the Nrf2 protein expression level in the nucleus was lower than that in HeLa-Scramble group (P < 0.01), suggesting that the Nrf2/ARE pathway can be further activated by WASH protein knockout.

Figure 7 Effect of JB concentration on HeLa-shWASHa cell-related proteins.

(A–F) The expression of the JB-induced bax, bcl-2, PCNA, WASH, and Nrf2/ARE signaling pathway proteins in cervical cancer cells after WASH protein sinking treatment for 12 h (**P < 0.01, *P < 0.05 vs control). Experiments were performed in triplicate, n = 3.

In addition, compared to the HeLa-Scramble, the amount expressed by the pro-apoptotic protein Bax was significantly increased in the HeLa-shWASH group. The expression of anti-apoptotic protein Bcl-2 was significantly decreased, and the results show that knockout WASH protein could further accelerate the apoptosis process of cancer cells. Compared with the HeLa-Scramble group, the expression level of cell cycle-related protein PCNA decreased, suggesting that DNA replication time may be prolonged after WASH protein is knocked out, and that the proliferation rate of cancer cells may be reduced. The results suggest that the WASH protein may be involved in JB regulation of cell cycle, antioxidant stress pathway, and apoptosis-related proteins PCNA, Nrf2, HO-1, Bax, and Bcl-2 to induce HeLa cell apoptosis and arrest cells in the G2/M phase.

JB regulates the immune function of cervical cancer cells and secreted cytokines

The effect of JB on the mRNA expression levels of IL-1α, IL-6, and IL-8 cytokines related to immune function of HeLa cells is shown in Fig. 8. Compared with the HeLa-Scramble group, the mRNA expression levels of these cytokines in the JB (0, 0.25, and 0.5 mM) groups significantly increased for 12 h (P < 0.01). JB regulated the mRNA expression levels of immune function-related cytokines in HeLa-Scramble and HeLa-shWASH cells in a concentration-dependent manner, with statistically significant differences between groups (P < 0.05). The results suggest that after the WASH protein was knocked out in cervical cancer cells. Over a range of concentrations, JB regulated mRNA expression levels of cytokines related to cellular immune function, such as IL-1α, IL-6, and IL-8, and increased the secretion of related cytokines, thereby contributing to cell inflammation and inducing apoptosis of HeLa cells.

Figure 8 Levels of IL-1α, IL-6, and IL-8 in each group detected by RT-qPCR.

The mRNA levels of IL-1α, IL-6, and IL-8 JB-induced in cervical cancer cells after WASH protein sinking treatment for 12 h (**P < 0.01, *P < 0.05 vs control). Experiments were performed in triplicate, n = 3.

Discussion

Cervical cancer is the fourth most common cancer in women and the seventh most common cancer worldwide, with 528,000 new cases globally and a projected 266,000 cervical cancer deaths in 2022, accounting for 7.5% of all female cancer deaths (Sung et al., 2021). The main treatments for cervical cancer are surgery, radiotherapy, chemotherapy, and immunotherapy. However, owing to the serious side effects of chemotherapy drugs, the therapeutic effect is significantly reduced, and various forms of drug resistance appear. Therefore, it is necessary to continue to search for new chemotherapeutic agents for cancer.

In recent years, owing to the effective therapeutic effect of botanical drugs on tumors, botanical drugs have gradually received increasing attention. Terpenoids are now regarded as potential natural anti-tumor drugs with chemotherapeutic and chemopreventative effects on many types of cancer. Compared with traditional chemotherapeutic drugs, natural chemotherapeutic drugs have better efficacy and fewer adverse reactions (Aiello et al., 2019). As an effective component of the natural plant medicine Euphorbia fischeriana, JB may play a significant role in the treatment and prevention of various types of cancer. It was reported that JB promoted Fas by activating apoptotic mitochondrial and death receptor pathways in MDA-MB-231 cells and U937 cells and upregulated the expression ratio of Bax/Bcl-2. The activation of FasL and Caspase-8 inhibited the proliferation of breast cancer MDA-MB-231 cells and U937 cells (Wang et al., 2011; Uto et al., 2012; Lin et al., 2012). Another in vivo study showed that morbidiolide B induces ROS-mediated apoptosis in bladder cancer cells by targeting thioredoxin and glutathione systems (Sang et al., 2021). Thus, several studies have shown that JB may be used for the treatment of malignant tumors in patients with cervical cancer. Using the WST-1 method, we investigated the influence of JB on HeLa vitality. The impact was found to be dose-dependent; HeLa growth was significantly inhibited as JB concentration increased.

Recent studies have found that changes in the skeletal structure and function of cells is also an essential factor leading to the occurrence and development of tumors. Many studies have shown that the malignant phenotype of tumor cells is usually associated with abnormal expression of actin-related proteins (Papakonstanti & Stournaras, 2008). Cytoskeletal regulatory protein WASH, also called WASHC1, is an important cytoskeletal regulatory protein of the Wiskott-Aldrich Syndrome Protein (WASP) family. However, the mechanism of cytoskeleton regulation in tumor cell migration and metastasis, as well as on tumor cell development remains unclear. Cytoskeletal regulatory proteins affect the expression of several important genes (Eberhardt et al., 2016), and the cytoskeleton is closely related to many signaling pathways (Gehren et al., 2015). These results suggest that WASH plays an important role in the occurrence and development of cervical cancer and is a potential prognostic target for cervical cancer with notable clinical significance. In addition, current data suggest that the presence or absence of WASH has a significant impact on sensitivity to JB; therefore, screening at the WASH level should be performed in cervical cancer patients before JB treatment is considered. It is suggested that analysis of WASH in other cervical cancer cell lines would be necessary to before its consideration for therapeutic purposes.

The Nrf2/ARE signaling pathway has become a new target for oxidative stress-related disease prevention and cancer therapy. After induced activation, Nrf2 is transferred into the nucleus and binds with antioxidant response element (ARE) to initiate the expression of downstream PHASE II metabolic enzymes and antioxidant enzymes, such as HO-1, NQO1, and SOD, to enhance the ability of cells to remove ROS, thus playing an anti-oxidative role (Bak et al., 2016; Petri, Körner & Kiaei, 2012; Calkins et al., 2009). Studies have found that the Nrf2/ARE signaling pathway mainly regulates the expression of anti-inflammatory genes and restrains inflammatory progression. Therefore, identification of new Nrf2-dependent anti-inflammatory phytochemicals has become an important point for drug discovery (Ahmed et al., 2017). This study focused on whether JB is involved in the regulation of the Nrf2/ARE signaling pathway, and if it affects the growth of HeLa cells. Compared with the negative control group, the expressions of Nrf2 and HO-1 proteins in HeLa cells after JB treatment were significantly increased, while the expressions of Nrf2 and HO-1 proteins in the HeLa-shWASH group after WASH protein knockout were lower than those in the HeLa-Scramble group. These results indicate that the WASH protein increased Nrf2 translocation into the nucleus and increased the expression of downstream target genes, such as HO-1. Therefore, we believe that the WASH protein can further promote JB regulation of the Nrf2/ARE signaling pathway and reduce the ability of cells to resist oxidative damage, promoting JB-induced mitochondrial damage in cervical cancer cells.

Cell proliferation and apoptosis rate are key indicators of cell viability and are very important for the occurrence and development of cancer. PCNA, Bax, and Bcl-2 are commonly used to observe cell proliferation and apoptosis, among which PCNA promotes cell growth and proliferation by enhancing the synthesis and extension of the DNA chain (Prelich et al., 1987). Bcl-2 and Bax are related to apoptosis, and the Bax/Bcl-2 ratio is an important factor determining apoptosis (Borujeni et al., 2016; Renault, Dejean & Manon, 2017). In this study, flow cytometry was used to study the effect of euphorbidiolide B on the cell cycle and apoptosis rate of HeLa cells. After treatment with different mass concentrations of euphorbidiolide B, the activity of HeLa cells was inhibited, and western blot results showed that the expression level of PCNA protein decreased significantly. The Bax/Bcl-2 ratio increased gradually compared with the control group. These results suggest that HeLa cells are trapped in the G2 phase, preventing damaged unreplicated DNA from entering mitosis, thereby ensuring cell integrity and accurate replication, avoiding excessive proliferation, and subsequently inducing apoptosis. In addition, flow cytometry results showed that euphorbidiolide B increased the apoptosis rate of HeLa cells in a dose-dependent manner, suggesting the induction of early apoptosis of HeLa cells.

Cancer cells themselves secrete cytokines related to inflammation, such as IL-1α, IL-6, and IL-8, which can activate macrophages. Activated macrophages further secrete substances that affect the development of cancer cells. IL-1α, IL-6, and IL-8 can promote immune processes involving the killing of tumor-related cells and induce apoptosis, which has the effect of directly inhibiting and killing tumor cells. IL-8 is an inflammatory factor mainly derived from mononuclear macrophages that can promote the growth and metastasis of tumor cells. In this study, the WASH protein effectively increased the levels of IL-1α, IL-6, and IL-8.

In addition, this experimental study found that Euphorbia fischeriana, a traditional Chinese herbal medicine, can be used in the treatment of various cancers. However, the sensitivity of Hela cells to JB (one of the effective ingredients of Euphorbia fischeriana) was lower (0.25 mM) than that to other botanical monomers, and further research on its treatment of cervical cancer is needed. In our earlier work, from flow cytometry analysis, a lower dose of JB (0.05, 0.1, or 0.2 mM) did not induce the extensive changes observed when applying a higher dose (0.25 or 0.5 mM). We did not identify any differentially expressed proteins that were associated with signal transduction upon application of a lower JB dose. Significant differences in cell cycle and cell death (G0-G1, S, and G2-M phases) were not observed during flow cytometry analysis.

The effects of various concentrations of chemical substances, such as propolin C, piceatannol, geraniol, â-ionone, and huanglian (an herbal extract) on the expression of different cyclins, CDKs, and other proteins in various cell lines had been investigated. In comparison with a higher dose of the various chemical substances, a lower dose of each in various treated cell lines was observed to have different biological activities (Chen, Wu & Lin, 2004; Li et al., 2000; Wolter et al., 2002; Duncan et al., 2004).

Conclusion

JB can inhibit proliferation and induce apoptosis of HeLa cells. The underlying mechanism may be related to WASH protein involvement in JB regulation of the Nrf2/ARE signaling pathway, which inhibits proliferation and promotes apoptosis of Hela cells. Further, JB induces the secretion of related cytokines (IL-1α, IL-6, and IL-8). Our results show that the WASH protein involved in the binding of JB to Nrf2 and HO-1 proteins. In addition, JB can also inhibit the metastasis and angiogenesis of cervical cancer cells in tumors, but the related mechanism requires verification. These results are expected to support the use of JB as a potential therapeutic mixture for cervical cancer. Further studies should be conducted to improve on our in vitro experiment and explore the underlying mechanism of JB in the treatment of cervical cancer.

Supplemental Information

Supplemental Information 1 Blots for all the 3 replicates and the results analysis.

Click here for additional data file.

Abbreviations

DMEM Dulbecco’s Modified Eagle Medium

PCNA proliferating cell nuclear antigen

PBS phosphate-buffered saline

JB Jolkinolide B

HPV human papillomavirus

Additional Information and Declarations

Competing Interests

Author Contributions

Data Availability

The authors declare that they have no competing interests.

Yu Hong conceived and designed the experiments, analyzed the data, prepared figures and/or tables, and approved the final draft.

Jicheng Liu conceived and designed the experiments, analyzed the data, prepared figures and/or tables, and approved the final draft.

Wanying Kong conceived and designed the experiments, performed the experiments, analyzed the data, prepared figures and/or tables, and approved the final draft.

Hui Li conceived and designed the experiments, performed the experiments, prepared figures and/or tables, and approved the final draft.

Ying Cui conceived and designed the experiments, performed the experiments, prepared figures and/or tables, and approved the final draft.

Yuchao Liu conceived and designed the experiments, performed the experiments, analyzed the data, authored or reviewed drafts of the article, and approved the final draft.

Zhihui Deng conceived and designed the experiments, analyzed the data, authored or reviewed drafts of the article, cell line, and approved the final draft.

Dezhi Ma conceived and designed the experiments, analyzed the data, authored or reviewed drafts of the article, and approved the final draft.

Keyong Zhang analyzed the data, authored or reviewed drafts of the article, and approved the final draft.

Jinghui Li conceived and designed the experiments, performed the experiments, analyzed the data, prepared figures and/or tables, authored or reviewed drafts of the article, and approved the final draft.

Minhui Li conceived and designed the experiments, prepared figures and/or tables, authored or reviewed drafts of the article, and approved the final draft.

The following information was supplied regarding data availability:

The raw data are available as Supplemental Files.

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
