# Peer review of "WASH regulates the oxidative stress Nrf2/ARE pathway to inhibit proliferation and promote apoptosis of HeLa cells under the action of Jolkinolide B"

_PeerJ, doi:10.7717/peerj.13499_

## Round 0.1 · original submission · Major Revisions

Dear Dr. Li,

Our reviewers have thoroughly reviewed your manuscript and have raised several important concerns. Address each comment by both the reviewers, especially focusing on following major concerns:

Methods not sufficiently explained, several labeling are not right, figure quality needs to be improved, correctness of claims in the introduction, spelling mistakes, explain the observations and interpretation in further detail, and provide the raw data where requested.

Reviewer 1 ·

Basic reporting

The authors have investigated the anti-tumor effects of JB at different concentrations on HeLa cell line as a model for cervical cancer. Although tumoricidal activity of JB has been demonstrated previously in other cell lines, the authors have attempted to test the impact of WASH protein expression on the effectiveness of JB. This aspect of the paper and the rationale behind testing the WASH mediated mechanism of JB action is poorly explained and needs further support. The authors have included certain references in discussion (lines 284-288), which would benefit from being moved to the introduction for clarification. The connection between WASH and Nrf2/ARE pathway also need further explanation and clarification (lines 297- 304 could be stated as a hypothesis in introduction). Language is professional and satisfactory but can be polished a bit more to enhance clarity of text.
The figures and tables in general need to be improved with respect to their labeling. Figures in the current version are extremely blurry and difficult to follow (labeling especially). Figure 1 does not add to the text or data in any way and as such can be deleted.
The authors have not explained their observations and interpretations in adequate detail. The results section needs to be expanded with additional comments on the impact of presence or absence of WASH protein in the two stable cell lines used. The authors also have frequently misinterpreted the results with no raw data/ values to support their conclusions. E.g. Fig3: authors have provided no explanation on how necrosis was quantified or make no comment on potential impact of WASH protein. Fig4: authors claim significant reduction in proliferation when the % decrease is less than 0.2% for either scramble or shWASHa groups and the error bars at time point 24hr are practically overlapping. Fig5: The flow plots are extremely poor resolution and make it impossible to judge the actual labels and data. Additionally, the authors show that BrDU positivity is greater in shWASH cells with increasing JB concentration but conclude that JB effectively suppresses proliferation in the absence of WASH protein which is the opposite of what the data suggest. Fig 6: Flow plots for panel c and g are practically identical and gating is inconsistent across the panels. This raises questions regarding the data validity. In many ways, the article fails to meet the standards and would need to be reassessed with clearer figures including raw data.

Experimental design

The authors need to improve explanation of the rationale behind (1) using only one cell line (2) rationale for testing impact of WASH protein and connection of WASH with Nrf2/ARE pathway (3) hypotheses that are being tested by each experiment. Certain methods used for assessing cell proliferation and apoptosis are redundant and do not add to the overall evidence presented. The authors have not described the experimental methods adequately and do not follow a universal method of describing cell densities across various set ups. E.g. line 139 states number of cells per well as 0.5x 10^5 which should be corrected to 5x10^4. Or on line 145, simply state the total number of cells per well as the concentration of cells is irrelevant to the set up. Line 141 only states that manufacturers protocols were followed with no details provided. Authors should add some basic information for each method performed in addition to stating manufacturer guidelines for reference. In the methods section, the authors have described using WST1 assay platform to measure cell viability but in discussion line 282, authors state the use of MTT which is a different assay- please clarify which assay was used. For statistical testing, the authors merely state the software used but not the actual statistical tests such as Student T test or ANNOVA etc. These details along with specific information regarding groups being tested is critical in evaluating the significance of the results.

Validity of the findings

The underlying raw data need to be provided by the authors to assess the validity of these findings. There is no clear indication or conclusion regarding the mechanistic impact of WASH protein on effect of JB and as such the manuscript fails to deliver data and conclusions that the experimental system was designed for.

Reviewer 2 ·

Basic reporting

The manuscript is clear and easy to read.
The introduction section needs to be revised as mentioned in comments.
WASH needs to be describe more in detail i.e. it function, relevance for cancer, protein structure/subunits.
The representation of data can be improved as stated in comments.
Rational for implicating WASH in JB sensitivity is not described.

Experimental design

Methods section needs improved description as described in comments.

Validity of the findings

The data are encouraging but are preliminary.
Data interpretation need to be revisited. Conclusions of the data need to be explained further.

Additional comments

In the manuscript submitted by Hong et. al., the authors report the effect of Jolkonilide B on HeLa cells. They report that WASH silenced cells have heightened cell death in response to JB. The data are encouraging, but there are several concerns as listed in Major and Minor concerns.
Major
1. The data are currently only performed on one cell line, HeLa. In order to extrapolate the results to cervical cancer, additional cell lines are needed. Also the current data indicates that WASH status determines JB sensitivity and therefore cervical cancers need to be screened for WASH levels before JB is considering it for treatment options.
2. Jolkinolide structure had been previously published in the literature and can inserted as a reference instead of representing as Fig1.
3. The authors use the name WASH in some instance and WASHa in other. Please provide the office NCBI Gene name and use that designation consistently throughout the manuscript. Also, it is suggested that the authors explain the rationale behind why they suspected that manipulation of WASH would sensitize cells to JB.
4. Please provide antibody details (company and catalog number) for WASH protein.
5. Check the figure label for Fig 3 a and d. They are control (non-JB) but the labels indicate JB.
6. The authors mention WASH protein sinking in the figures and legends. It is suggested to describe this in the methods section.
7. For better representation of the data in Figure 4, normalize the data from time 0 of all mentioned concentration to 100% and indicate % cell viability of the different treatment conditions.
8. In Figure 5 the authors use BrdU incorporation to assess proliferation. BrdU is incorporated into dividing cells. The authors previously show more cell death shWASH in 0.5mM of JB than 0.25mM. BrdU appears to be higher in shWASH in 0.5mM of JB which is not consistent with other reported data/images of cell death. Also shControl cells have lesser BrdU incorporation than treated cells. Please explain.
9. Cell cycle data also indicates increased cells in S phase of cell cycle in JB treated cells. This is consistent with the BrdU data. The authors are suggested to explain these observations.
10. The cytokine profiling data indicated increased expression of IL1α, IL-6 and IL-8 at basal level in shWASH cells and is not statistically higher upon JB treatment of these cells. Further IL-6 increases only in 0.25mM but not 0.5mM. This data indicates that WASH silencing regulates cytokine gene expression and the results must be revised to indicate this rather than stating that JB influences cytokine and inflammation/immune cell function.
Minor
1. For technical correctness, please use the format “shWASH” instead of ShWash/ShWasha throughout the manuscript.
2. revise introduction for accuracy. The authors state Cancer is one of the deadliest diseases and cause for highest mortality rates for humans. This statement is not correct.
“The top global causes of death, in order of total number of lives lost, are associated with three broad topics: cardiovascular (ischaemic heart disease, stroke), respiratory (chronic obstructive pulmonary disease, lower respiratory infections)” Ref:
https://www.who.int/news-room/fact-sheets/detail/the-top-10-causes-of-death
3. Replace purinomycin to puromycin in Ln 110

---

## Round 0.2 · Minor Revisions

Dear Dr. Li,

The reviewer has couple of minor concerns which need to be addressed.

Reviewer 2 ·

Basic reporting

Please see additional comment

Experimental design

Please see additional comment

Validity of the findings

Please see additional comment

Additional comments

Regarding the response to #1 Major concern, it is suggested that the authors include a line in the discussion to mention that analysis of WASH in other cervical cancer cell lines would be necessary to before its consideration for therapeutic purposes.

With respect to #3, since the official gene ID lists WASHC1 as the official symbol with alternate names of WASH1 and WASH, it is necessary to list all these three versions next to the gene ID for clarity.

Please remove the word “artificial” to describe HeLa cell line in Methods section 2.4

All other concerns have been addressed.

---

## Round 0.3 · Minor Revisions

Congratulations Dr. Li,

Based on the reviews I received, before accepting the manuscript we would require following modifications.

1. Currently blots are provided for only one experiment. Submit blots for all the 3 replicates, along with the spreadsheet of the results analysis.

2. Remove the RAR files and replace them with either uncompressed data files in their native format or as a more accessible compressed file format (such as ZIP or 7Z ).

---

## Round 0.4 · accepted · Accept

Dear Dr. Li,

Thank you for uploading the necessary files.